# Simulation of Aerosol Evolution within Background Pollution for Nucleated Vehicle Exhaust via TEMOM

Can Tu [1], Yueyan Liu [2,*], Taiquan Wu [2] and Mingzhou Yu [1]

1   Laboratory of Areosol Science and Technology, China Jiliang University, Hangzhou 310018, China; s1701081110@cjlu.edu.cn (C.T.); mzyu@cjlu.edu.cn (M.Y.)
2   College of Modern Science and Technology, China Jiliang University, Yiwu 322000, China; 07a0803079@cjlu.edu.cn
*   Correspondence: 12a1803148@cjlu.edu.cn

**Abstract:** This work is intended to study the effect of background particles on vehicle emissions in representative realistic atmospheric environments. The coupling of Reynolds-Averaged Navier–Stokes equation (RANS) and Taylor-series Expansion Method Of Moments (TEMOM) is performed to track the emissions of the vehicle and simulating the evolution of the matters. The transport equation of mass, momentum, heat, and the first three orders of moments are taken into account with the effect of binary homogeneous nucleation, Brownian coagulation, condensation, and thermophoresis. The parameterization model is utilized for nucleation. The measured data for Beijing's particle size distribution under both polluted and nonpolluted conditions are utilized as background particles. The relationship between the macroscopic measurement results and the microscopic dynamic process is analyzed by comparing the variation trend of several physical quantities in the process of aerosol evolution. It is found with an increase of background particle concentration, the nucleation is inhibited, which is consistent with the existing studies.

**Keywords:** TEMOM; RANS; background aerosol; vehicle emissions

## 1. Introduction

The health effects of particulate matter (PM) air pollution have long been a matter of widespread concern [1–4]. Much of this work focuses on measurements of aerosols at different times of the day. This method can identify periods when particle concentrations are highest directly to judge the degree of its impact on health. However, this research method cannot analyze the factors affecting the evolution of aerosols and the origin of aerosols. The reason is that the pollution sources, temperature, humidity, etc., change with time during the day. All of these changes affect the evolution of aerosols as the aerosol evolves over the day. Therefore, the factors cannot be decoupled. Therefore, simulation plays an essential role in solving how physical quantities affect the evolution of aerosols. Simulation can keep irrelevant variables unchanged and only change the target variable.

The coupling of fine particles and macroscopic flow field belongs to multiphase flow in fluid mechanics. In this study, there are only two phases: air and particles. When dealing with the multiphase flow, the continuous phase is generally described by Navier–Stokes equation, and there are many options for the discrete phase. There are generally three choices. First, the discrete phase is equivalent to the continuous phase, which is the same as the continuous phase and is described by the NS equation. Second, use Lagrangian particles for tracking; third, use the PBE (Population Balance Equation) equation to describe the discrete phase. We are using the third method here.

Silva's work [5] has shown that these particulate matters, especially with diameters below 100 nm, have a more adverse influence on human health than ultrafine and coarse particles. The first study found the effect on background aerosol, including coagulation, condensation, and nucleation, coupled with Navier–Stokes equation. By comparing with

the simulation data, the effect of background aerosol on the evolution process is investigated. We have set up six cases to evaluate the effect of background pollution on nucleation. The simulation method had been proved with a good agreement by Yu [6].

Many studies have reported aerosol's evolution under pollution conditions [7–9]. The background of pollution has a significant impact on the evolution of aerosols, which has reached a consensus. However, these studies are mainly based on experiments and measurements. They do not involve the quantitative analysis of the aerosol evolution process of the background physical quantities of the pollution background.

Kulmala [10] uses the thermodynamically consistent theory to treat the binary nucleation rate as a function of temperature, relative humidity, and acidity. Compared with the classical approach, they increased the computational efficiency of simulated nucleation by 25 times. Then, Vehkamaki [11] extended the applicable temperature range of the work to the high-temperature range, which led to applying this research to simulate vehicle emissions. The evolution of vehicle emissions will be diluted in three-dimensional space due to the consideration of the flow field. The temperature, concentration, etc. will change accordingly. This can lead to significant changes in particle size and its distribution. This is also the reason why the Euler–Euler model is challenging to track the particle phase. It is complicated to consider multiple processes at the same time. Thus, researchers have separately studied the influence of a single process on the single properties of particle population. For example, Kim [12,13] studied the effect of environmental dilution on the particle size distribution and concentration changes of the carrier by taking the particle size distribution as a function of time. Tat Leung Chan studied the influence of turbulent kinetic energy on coagulation [14], dilution on number concentration, and the relationship between internal combustion engine operating state and particle size [15].

The influence of the vehicle's shape on the flow field cannot be ignored during the driving process of the vehicle. The enrichment and desalination of the background particle concentration by the flow field will indirectly affect the evolution of aerosols. As Figure 2 shows, the figure uses log scale when mapping data to colors to observe the nucleation rate with a large span of magnitude. The vehicle's shape limits the spatial range of the evolution of particulate matter in the rear area of the car. This interference will increase turbulence and indirectly affect the coagulation process [16]. There are also related experiments to study the influence of the airflow angle of the exhaust tailpipe on the evolution of aerosols [17].

The study shows that the main source of urban pollution is automobile emission. However, the research on emission evolution focuses on the evolution of emission in a pure environment. In practice, a considerable degree of background particulate matter already exists in urban atmospheres. Moreover, the characteristics of background particulate matter under the condition of pollution are very different from those under the condition of no pollution. In practice, Seipenbusch [18] found that background particulate matter has its parameter characteristics. We believe that this has a significant impact on the evolution of emissions. To test this idea, Seipenbusch [18] analyzed fully evolved aerosols.

We plan to use simulation methods to predict the evolution of aerosols in the presence of background particles in order to find out the law of the control of urban air pollution and other occasions of aerosol physical property control.

## 2. Governing Equation

### 2.1. CFD Model

The one-way couple was used between the continuous and discrete phase considering that the particle has a minimum mass. Thus, we solved the fluid dynamic equation and moment transport equation separately. To obtain the information of fluid flow, the RANS equation coupled with Renormalized Group (RNG) $k \sim \omega$ turbulent model is utilized. The governing equations are described as follows:

$$\frac{\partial \overline{u_i}}{\partial x_i} = 0$$

$$\rho \frac{\partial \overline{u_i}}{\partial t} + \frac{\partial}{\partial x_j}\left(\rho \overline{u_i u_j}\right) = -\frac{\partial \overline{p}}{x_i} + \mu \frac{\partial^2 \overline{u_i}}{\partial x_j \partial x_i} - \frac{\partial R_{ij}}{\partial x_j} + \overline{f_i}$$

(1)

in which

$$R_{ij} = \mu_t \left(\frac{\partial \overline{u_i}}{\partial x_j} + \frac{\partial \overline{u_j}}{\partial x_i}\right) - \frac{2}{3}\rho k \delta_{ij}$$

(2)

where $\overline{u_i}$ and $\overline{u_j}$ are mean velocity in $i$ th and $j$th direction coordinates, respectively; $\rho$ is the fluid density; $\mu$ is the laminar viscosity; $\overline{f_i}$ is the body force in the $i$th direction coordinate, and $\mu_t$ is the turbulent dynamic viscosity. The enthalpy $h$ is calculated by solving the following energy equation in this study:

$$\frac{\partial (\rho h)}{\partial t} + \frac{\partial}{\partial x_i}(\rho h \overline{u_i}) = \frac{\partial}{\partial x_i}\left(\frac{k_t}{C_p}\frac{\partial h}{\partial x_i}\right)$$

(3)

where $k_t$ is the thermal conductivity and $C_p$ is the specific heat at constant pressure. The temperature $T$ is obtained by:

$$h = C_p T$$

(4)

the turbulence kinetic energy $k_t$ and its rate of dissipation $C_p$ are obtained by the equations below:

$$\frac{\partial (\rho k)}{\partial t} + \frac{\partial}{\partial x_i}(\rho k u_i) = \frac{\partial}{\partial x_j}\left[\left(\mu + \frac{\mu_\tau}{\delta_k}\right)\frac{\partial k}{\partial x_j}\right] + P_k + P_b - \rho\epsilon - Y_M$$

(5)

$$\frac{\partial (\rho \epsilon)}{\partial t} + \frac{\partial}{\partial x_i}(\rho \epsilon u_i) = \frac{\partial}{\partial x_j}\left[\left(\mu + \frac{\mu_\tau}{\delta_\epsilon}\right)\frac{\partial \epsilon}{\partial x_j}\right] + c_{\epsilon 1}\frac{\epsilon}{k}(P_k + c_{\epsilon 3}P_b) - c_{\epsilon 2}\rho\frac{\epsilon^2}{k}$$

(6)

where $P_k$ represents the production of turbulence kinetic energy due to the mean velocity gradients. $P_b$ represents the production of turbulence kinetic energy due to buoyancy. $Y_M$ represents the partition of fluctuating dilatation incompressible turbulence to the overall dissipation rate. The turbulent Prandtl number for $k$, $\delta_k$, is 1; the turbulent Prandtl number for $\epsilon$, $\delta_\epsilon$, is 1.30; and $c_{\epsilon 1}$ is 1.44, $c_{\epsilon 2}$ is 1.92; the turbulent viscosity, $\mu_t$, is obtained by:

$$\mu_t = c\rho\vartheta\ell = \rho c_\mu \frac{k^2}{\epsilon}$$

(7)

where $c_\mu$ is 0.09, $\vartheta = k^{1/2}$, $\ell = k^{3/2}/\epsilon$; and energy Prandtl number and Prandtl number are 0.85.

### 2.2. PBE Solution

Several studies has shown how the mechanism of PM evolution is established. It is only since Smoluchowski's work [19] that the study of coagulation kinetics has gained momentum. The work describes monodisperse systems of spherical particles. Systems are complex in most cases, in which particles are not the same size. Muller [20] expanded the Smoluchowski theory to polydisperse systems.

Population balance equation (PBE), as an extension of the Smoluchowski equation, is used to describe the time evolution of particle size distributions. To date, several numerical solutions for the PBE had been developed, including moment method (MM) [21], sectional method (SM) [22], and the Monte Carlo method (MC) [23]. The above three methods can be divided into sub-methods. We used TEMOM (Taylor-series Expansion Method Of Moments) [24] in this study, a type of method of moments. The present study concerns the new particle formation by binary homogeneous nucleation in a given aerosol situation and

growth by coagulation and condensation. Thus, the three source terms are described in PBE as follows:

$$\frac{\partial n(v,t)}{\partial t} = \left\{ \frac{1}{2} \int_{v_{\min}}^{v_{\max}} \beta(v-u,u,t)n(v-u,t)n(u,t)\mathrm{d}u - n(v,t)\int_{v_{\min}}^{v_{\max}} \beta(v,u,t)n(u,t)\mathrm{d}u \right\}_{\text{coagulation}} \tag{8}$$

$$- \left\{ \frac{\partial(I(v,t)n(v,t))}{\partial v} \right\}_{\text{condensation}} + \{J(v,t)\delta(v_{\text{nul}},v)\}_{\text{nucleation}}$$

where $n(v,t)$ is number density function; $n(v,t)\mathrm{d}v$ is the number density of particles size between $v$ and $v + \mathrm{d}v$ at the time $t$; $\beta(v,u,t)$ is the collision kernel for particles of volume $v$ and $u$ at the time $t$; $I(v,t)$ is the condensation kernel for particles of volume $v$ at the time $t$; and $J(v,t)$ is the nucleation kernel for particles of volume $v$ at the time $t$.

### 2.3. Coagulation Kernel

As summarized in the review [25], the vehicle nucleated particles are primarily between 32 nm and 90 nm. Thus, the collision kernel $\beta$ in the present study is limited in the free molecular regime in the present study [21]:

$$\beta(v,u) = B_1\left(\frac{1}{v} + \frac{1}{u}\right)^{1/2}(v^{1/3} + u^{1/3})^2 \tag{9}$$

in which

$$B_1 = \left(\frac{3\pi}{4}\right)^{1/6}\left(\frac{6k_bT}{\rho_p}\right)^{1/2} \tag{10}$$

where $k_b$ is Boltzmann constant; and $\rho_p$ is particles density. The $k$th moment of particle size distribution is defined as follows:

$$M_k(t) = \int_0^\infty v^k n(v,t)\mathrm{d}v \tag{11}$$

If only the first three moments are considered, the coagulation kernel moment equation is obtained by multiplying Equation (8) by $v^k$ and then integrating it from 0 to $\infty$ [24]:

$$\left(\frac{\partial m_0}{\partial t}\right)_{\text{coa}} = \frac{\sqrt{2}B_2(65m_2^2 m_0^{23/6} - 1210m_2 m_1^2 m_0^{17/6} - 9223m_1^4 m_0^{11/6})}{5184m_1^{23/6}}$$

$$\left(\frac{\partial m_1}{\partial t}\right)_{\text{coa}} = 0 \tag{12}$$

$$\left(\frac{\partial m_2}{\partial t}\right)_{\text{coa}} = \frac{\sqrt{2}B_2(701m_2^2 m_0^{11/6} - 4210m_2 m_1^2 m_0^{5/6} - 6859m_1^4 m_0^{-1/6})}{2592m_1^{11/6}}$$

TEMOM is a general method for solving the PBE. The closure of the moment equations is approached by the Taylor-series expansion technique. This is why this method has no prior requirement for particle size distribution. As the study [24] shows, 3-order TEMOM is preferable when precision and efficiency are simultaneously considered in which the first three order moments are calculated. In other words, the most important indexes for describing aerosol, including particle number density, particle mass, and geometric standard deviation, are obtained.

### 2.4. Condensation Kernel

For particles smaller than the mean free path of the surrounding gas, the condensation models for sulfuric acid vapor molecules is [26]:

$$I(v,t)\mathrm{d}t = \frac{\pi d_p^2 v_m(p_1 - p_d)}{(2\pi m k_b T)^{1/2}} \tag{13}$$

where $p_1$ is the partial pressure of sulfuric acid vapor; $p_d$ is the partial pressure of $H_2SO_4$ vapor at the particle surface, and $v_m$ is the volume of one $H_2SO_4$ molecule.

Because the water vapor concentration is very high compared with sulfuric acid vapor and pre-existing particle concentration, the new solution particle is quickly in equilibrium with the surrounding water vapor [27]. Therefore, the molar fraction of sulfuric acid in particles is approximately equal to the molar fraction in gas phases. Taking into account the effect of water vapor and hydrate on condensation, the condensation size growth rate $\theta$ is:

$$\theta = \frac{dv_c}{dt} \alpha \tag{14}$$

where $\alpha$ is a coefficient that makes $\theta$ represent the volume growth rate of both sulfuric acid and water molecules by the volume growth rate of sulfuric acid molecules, in which $\alpha \times v_a$ can be interpreted as the average volume of molecules of sulfuric acid and molecules of water or the equivalent number of sulfuric acid molecules. Here, $v_a$ is the volume of the sulfuric acid molecules. $\alpha$ is [6]:

$$\alpha = 1 + \frac{(1 - \chi)v_w}{\chi v_a} \tag{15}$$

where $\chi$ is the mole fraction of sulfuric acid vapor; $v_w$ is the volume of the water molecule. Substitute $p_1 = Y_1 k_b T$, $d_p = ((6v)/\pi)^{1/3}$ and Equation (13) into Equation (14). The condensation growth rate $\theta$ is:

$$\theta = B_2 v^{2/3} \eta \alpha \tag{16}$$

where $B_2 = (36\pi)^{1/3} n_s (k_b T/2\pi m_1)^{1/2} v_m$, $\eta = Y_1/n_s$ and $n_s$ is the reference sulfuric acid concentration.

### 2.5. Thermophoresis and Particle Diffusion

In the present study, particle diameters are usually smaller than the mean free path of the gas. The velocity of thermophoresis $u_{th}$ in Fredlander' work [28] is

$$u_{th} = \frac{-0.55\mu \nabla T}{\rho_g T} \tag{17}$$

The particle diffusion coefficient is the sum of $\Gamma_t$ and $\Gamma_B$. Here, $\Gamma_t$ is the turbulent diffusivity and $\Gamma_B$ is the Brownian diffusivity [29].

$$\Gamma_B = k_b T \frac{C_c}{3\pi\mu d_a} \tag{18}$$

where $C_c$ is the Cunningham correction factor [30], and $d_a$ is the volume-averaged particle diameter.

### 2.6. Moment Transport Equation

The moment transport equation for the $k$th moment based on the TEMOM is [24]:

$$\frac{\partial m_k}{\partial t} = -\frac{\partial (u_j + u_{(th)j})m_k}{\partial x_j} + \frac{\partial}{\partial x_j}\left(\Gamma \frac{\partial m_k}{\partial x_j}\right)$$
$$+ kB_1\eta m_{k-1/3}\alpha + J(v^*)v^{*k} + \left(\frac{\partial m_k}{\partial t}\right)_{coa} \tag{19}$$

When three source terms can be considered at the same time, the Equation (19) expands to:

$$\frac{\partial m_0}{\partial t} = J^* + \frac{\sqrt{2}B_2(65m_2^2m_0^{23/6} - 1210m_2m_1^2m_0^{17/6} - 9223m_1^4m_0^{11/6})}{5184m_1^{23/6}}$$

$$\frac{\partial m_1}{\partial t} = J^*\frac{v^*}{v} + m_{2/3}\eta\alpha \tag{20}$$

$$\frac{\partial m_2}{\partial t} = J^*\left(\frac{v^*}{v}\right)^2 + 2m_{5/3}\eta\alpha - \frac{\sqrt{2}B_2(701m_2^2m_0^{11/6} - 4210m_2m_1^2m_0^{5/6} - 6859m_1^4m_0^{-1/6})}{2592m_1^{11/6}}$$

The dimensionless calculation equation is used in the process. The characteristic timescale for particle growth is calculated by $\tau = [n_s s_1(k_b T/2\pi m_1)^{1/2}]^{-1}$. Dimensionless time is calculated by $\theta = t/\tau$. Kth moment is calculated by $m_k = M_k/n_s v_1^k$. Nucleation rate is calculated by $J^* = J/(n_s/\tau)$. Number concentration is calculated by $S = n_1/n_s$. Rate of gas to particle production is calculated by $R^* = R\tau/n_s$.

## 3. Simulation Configuration

### 3.1. Flow Field

It is necessary to pay special attention to the engine exhaust plume rather than the whole field where the engine truck is located because the time and space scales of nucleation and growth processes are tiny. The simulation flow field is shown in Figure 1. The two-dimensional space is a plane with a length of 10 m and a width of 3 m.

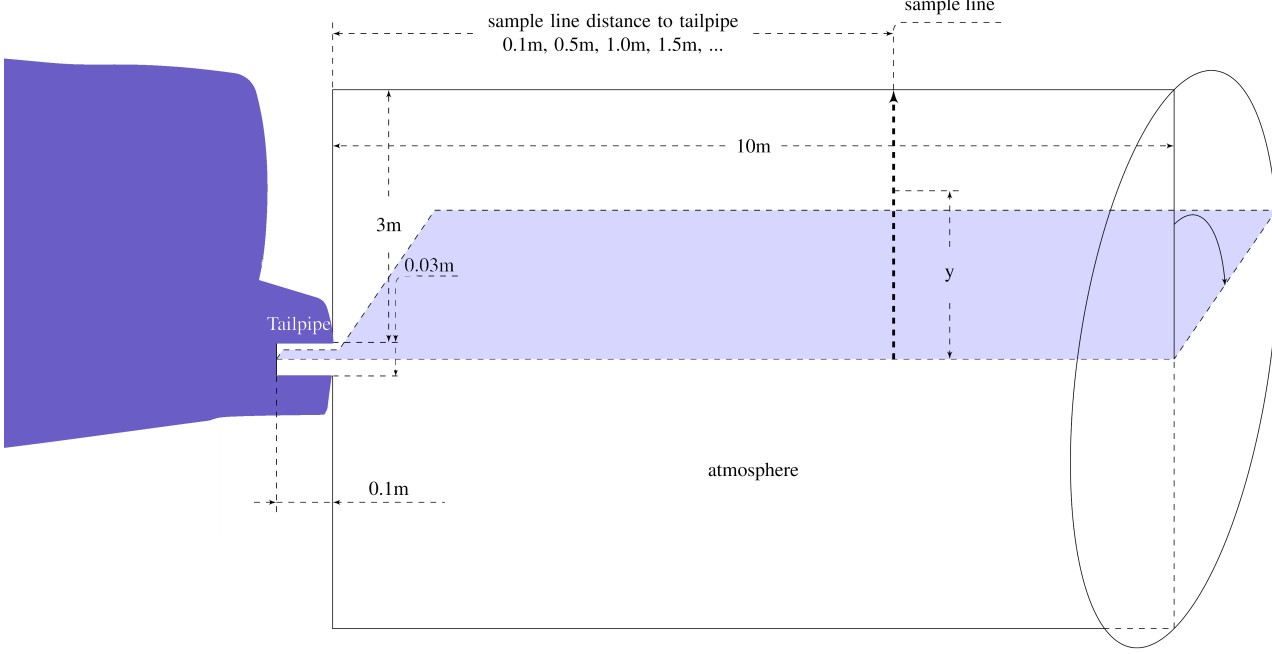

**Figure 1.** Flow field of simulation.

The mesh used in this paper is a square with a side length of 1 mm in the plume area (30 cm × 90 cm). In other regions, the edge length of the mesh increases by 1.01 times. The edge length of the farthest mesh is 20 mm. According to our calculation experience, the density of the mesh far exceeds the demand.

To study the details of the influence of various physical quantities on the evolution, we only consider the evolution of exhaust gas in the background pollution without involving experiments. In research similar to the present study [6], its simulation situation is similar to that of Ning's experiment [17]. The research proved the effectiveness of this method. The difference between this study and Yu's study lies in background pollutants and the absence of ground in the flow field. At the same time, this work applies to three-dimensional space.

Five scalars are defined in the flow field, representing water vapor, sulfuric acid vapor, and third-order moments, respectively. The flow, particle, and temperature are developed numerical code in SIMPLEC (Semi-Implicit Method for Pressure-linked Equations-Consistent).

### 3.2. Initial Conditions Calculation

The initial boundary condition numerical is calculated from research on the atmosphere of Beijing [31]. Table 1 is the atmospheric background aerosol data under pollution and non-pollution conditions in Beijing over three years.

**Table 1.** Average lognormal fit parameters of the particle number size distributions and meteorological factors on polluted and nonpolluted days in the summers of 2004, 2005, and 2006.

| Mode | Nucleation | | | Aitken | | | Accumulation | | |
|---|---|---|---|---|---|---|---|---|---|
| | Ni | GMD | $\delta$ | Ni | GMD | $\delta$ | Ni | GMD | $\delta$ |
| 2004 nonpolluted | 16 | 15.5 | 1.80 | 27 | 60.4 | 1.87 | 3 | 200 | 1.70 |
| 2004 polluted | 14 | 17.0 | 1.90 | 19 | 80.9 | 1.93 | 7 | 245 | 1.58 |
| 2005 nonpolluted | 7 | 14.4 | 2.00 | 20 | 58.9 | 2.00 | 5 | 174 | 1.71 |
| 2005 polluted | 6 | 18.0 | 2.00 | 17 | 75.8 | 1.99 | 8 | 251 | 1.59 |
| 2006 nonpolluted | 9 | 14.3 | 1.92 | 15 | 61.8 | 2.00 | 2 | 225 | 1.64 |
| 2006 polluted | 5 | 18.7 | 1.89 | 15 | 84.6 | 1.96 | 6 | 246 | 1.54 |

Where Ni is mode particle number concentratio ($\times 10^3$ cm$^{-3}$); GMD is geometric mean diameter of the mode (nm); and $\delta$ is standard deviation of the mode.

The first three initial moments were calculated by Equation (21):

$$M_m = N V_g^m \exp\left( m^2 \frac{w_g^2}{2} \right) \tag{21}$$

where $w_g = 3 \ln \delta_g$. The calculation results are Table 2.

**Table 2.** Initial moments of 2004, 2005, and 2006.

| Condition | | Nucleation | Aitken | Accumulation | Sum | Non-Dimensional |
|---|---|---|---|---|---|---|
| Nonpolluted 2004 | | $1.60 \times 10^{10}$ | $2.70 \times 10^{10}$ | $3.00 \times 10^{09}$ | $4.60 \times 10^{10}$ | $3.07 \times 10^{00}$ |
| Polluted 2004 | | $1.40 \times 10^{10}$ | $1.90 \times 10^{10}$ | $7.00 \times 10^{09}$ | $4.00 \times 10^{10}$ | $2.67 \times 10^{00}$ |
| Nonpolluted 2005 | | $7.00 \times 10^{09}$ | $2.00 \times 10^{10}$ | $5.00 \times 10^{09}$ | $3.20 \times 10^{10}$ | $2.13 \times 10^{00}$ |
| Polluted 2005 | $m_0$ | $6.00 \times 10^{09}$ | $1.70 \times 10^{10}$ | $8.00 \times 10^{09}$ | $3.10 \times 10^{10}$ | $2.07 \times 10^{00}$ |
| Nonpolluted 2006 | | $9.00 \times 10^{09}$ | $1.50 \times 10^{10}$ | $2.00 \times 10^{09}$ | $2.60 \times 10^{10}$ | $1.73 \times 10^{00}$ |
| Polluted 2006 | | $5.00 \times 10^{09}$ | $1.50 \times 10^{10}$ | $6.00 \times 10^{09}$ | $2.60 \times 10^{10}$ | $1.73 \times 10^{00}$ |
| Nonpolluted 2004 | | $1.48 \times 10^{-13}$ | $1.82 \times 10^{-11}$ | $4.46 \times 10^{-11}$ | $6.29 \times 10^{-11}$ | $3.65 \times 10^{00}$ |
| Polluted 2004 | | $2.30 \times 10^{-13}$ | $3.69 \times 10^{-11}$ | $1.38 \times 10^{-10}$ | $1.75 \times 10^{-10}$ | $1.02 \times 10^{01}$ |
| Nonpolluted 2005 | | $9.51 \times 10^{-14}$ | $1.86 \times 10^{-11}$ | $5.04 \times 10^{-11}$ | $6.91 \times 10^{-11}$ | $4.00 \times 10^{00}$ |
| Polluted 2005 | $m_1$ | $1.59 \times 10^{-13}$ | $3.27 \times 10^{-11}$ | $1.74 \times 10^{-10}$ | $2.07 \times 10^{-10}$ | $1.20 \times 10^{01}$ |
| Nonpolluted 2006 | | $9.35 \times 10^{-14}$ | $1.61 \times 10^{-11}$ | $3.59 \times 10^{-11}$ | $5.21 \times 10^{-11}$ | $3.02 \times 10^{00}$ |
| Polluted 2006 | | $1.06 \times 10^{-13}$ | $3.65 \times 10^{-11}$ | $1.07 \times 10^{-10}$ | $1.44 \times 10^{-10}$ | $8.32 \times 10^{00}$ |
| Nonpolluted 2004 | | $3.05 \times 10^{-35}$ | $4.15 \times 10^{-31}$ | $8.36 \times 10^{-30}$ | $8.78 \times 10^{-30}$ | $4.42 \times 10^{02}$ |
| Polluted 2004 | | $1.54 \times 10^{-34}$ | $3.50 \times 10^{-30}$ | $1.79 \times 10^{-29}$ | $2.14 \times 10^{-29}$ | $1.08 \times 10^{03}$ |
| Nonpolluted 2005 | | $9.75 \times 10^{-35}$ | $1.30 \times 10^{-30}$ | $6.77 \times 10^{-30}$ | $8.07 \times 10^{-30}$ | $4.07 \times 10^{02}$ |
| Polluted 2005 | $m_2$ | $3.19 \times 10^{-34}$ | $4.45 \times 10^{-30}$ | $2.63 \times 10^{-29}$ | $3.08 \times 10^{-29}$ | $1.55 \times 10^{03}$ |
| Nonpolluted 2006 | | $4.47 \times 10^{-35}$ | $1.31 \times 10^{-30}$ | $5.82 \times 10^{-30}$ | $7.13 \times 10^{-30}$ | $3.59 \times 10^{02}$ |
| Polluted 2006 | | $8.63 \times 10^{-35}$ | $5.23 \times 10^{-30}$ | $1.02 \times 10^{-29}$ | $1.54 \times 10^{-29}$ | $7.77 \times 10^{02}$ |
| Nonpolluted 2004 | | $4.98 \times 10^{-06}$ | $1.40 \times 10^{-04}$ | $1.37 \times 10^{-04}$ | $3.39 \times 10^{-04}$ | $6.52 \times 10^{-02}$ |
| Polluted 2004 | | $5.99 \times 10^{-06}$ | $1.92 \times 10^{-04}$ | $4.15 \times 10^{-04}$ | $7.40 \times 10^{-04}$ | $1.42 \times 10^{-01}$ |
| Nonpolluted 2005 | | $2.46 \times 10^{-06}$ | $1.18 \times 10^{-04}$ | $1.75 \times 10^{-04}$ | $3.43 \times 10^{-04}$ | $6.60 \times 10^{-02}$ |
| Polluted 2005 | $m_{2/3}$ | $3.30 \times 10^{-06}$ | $1.64 \times 10^{-04}$ | $5.03 \times 10^{-04}$ | $7.79 \times 10^{-04}$ | $1.50 \times 10^{-01}$ |
| Nonpolluted 2006 | | $2.80 \times 10^{-06}$ | $9.73 \times 10^{-05}$ | $1.07 \times 10^{-04}$ | $2.58 \times 10^{-04}$ | $4.97 \times 10^{-02}$ |
| Polluted 2006 | | $2.55 \times 10^{-06}$ | $1.73 \times 10^{-04}$ | $1.02 \times 10^{-29}$ | $1.54 \times 10^{-29}$ | $7.77 \times 10^{02}$ |

There has been a dimensionless treatment in Section 2.6 Dimensionless processing is used to compare the calculation accuracy between different mathematical models without considering the influence of the number and volume of particles. In this study, in the simulation process, the particle size span is large. The effective number of digits is limited, and the rounding error in the calculation process, so the calculation also uses dimensionless processing.

### 3.3. Boundary Setting

The evolution of aerosols under six background particles with different size distributions was simulated using the data in Table 2 as the initial moment value, the evolution of aerosols under six background particles was simulated. The upper and lower boundaries are fixed values, meaning that the environment is maintained. When the moment value disappears during the simulation process, the environment can be supplemented; when the moment value increases sharply, the environment can be diluted. Only the inner wall of the exhaust pipe does not increase or decrease the moment—so zero flux. Table 3 is an example of the boundary conditions under the condition that there is no pollution in the simulation in 2004. S. means scalar.

**Table 3.** Example boundary setting of nonpolluted 2004.

| Boundary | Temperature (K) | Velocity (m/s) | Boundary Type | Boundary Value | | | | |
|---|---|---|---|---|---|---|---|---|
| | | | | S.1 | S.2 | S.3 | S.4 | S.5 |
| inlet | 400 | 4.8 | Fixed value | 1 | 1000 | 0 | 0 | 0 |
| inlet wall | 350 | | Fixed flux | 0 | 0 | 0 | 0 | 0 |
| right | 300 | | Fixed value | 0.3 | 0 | $3.07 \times 10^{00}$ | $3.65 \times 10^{00}$ | $4.42 \times 10^{02}$ |
| up (down) | 300 | | Fixed value | 0.3 | 0 | $3.07 \times 10^{00}$ | $3.65 \times 10^{00}$ | $4.42 \times 10^{02}$ |
| left | 300 | | Fixed value | 0.3 | 0 | $3.07 \times 10^{00}$ | $3.65 \times 10^{00}$ | $4.42 \times 10^{02}$ |

### 4. Results and Discussion

The simulation results are shown through the sampling line shown in Figure 1. In the plane, the data are taken at a certain distance from the exhaust pipe, 0.1 m, 0.5 m, 2.0 m, 5.0 m. The nucleation rate has no value beyond the length greater than 1 m, so the sampling line distance of the nucleation rate is 0.1 m, 0.25 m, 0.5 m, 0.75 m. In the calculation process of nucleation rate, the upper and lower limits of nucleation rate are limited in order to ensure the correct value. The nucleation rate generally cannot break the upper limit. The lower limit can be considered to be approximately zero due to a difference of about 15 orders of magnitude from the normal value. The most detailed account of the process is to be found in the work of Binder and Stauffer [32].

Take the flow field without pollution in 2004 as an example, as shown in Figure 2, the speed of the flow field, $m_0$, $m_1$, and nucleation rate are shown in Figure 2. This study uses three years of data, and there are two situations each year. Six conditions need to be analyzed in one simulation. There are eight sampling lines in each situation, and each sampling line has five custom scalar quantities and eight custom physical quantities. There are 312 ($6 \times 4 \times 13$) cases of data that need to be classified and displayed. Quantitative analysis in two-dimensional figures is complicated. Other data will be organized and displayed according to conditions to facilitate the comparison of differences.

In the following figure, the blue dotted line represents the data under the condition of no pollution, and the solid red line represents the data under the condition of pollution. The horizontal $y$-axis represents the vertical distance from the exhaust pipe on the sampling line, and the maximum is 3.15 m.

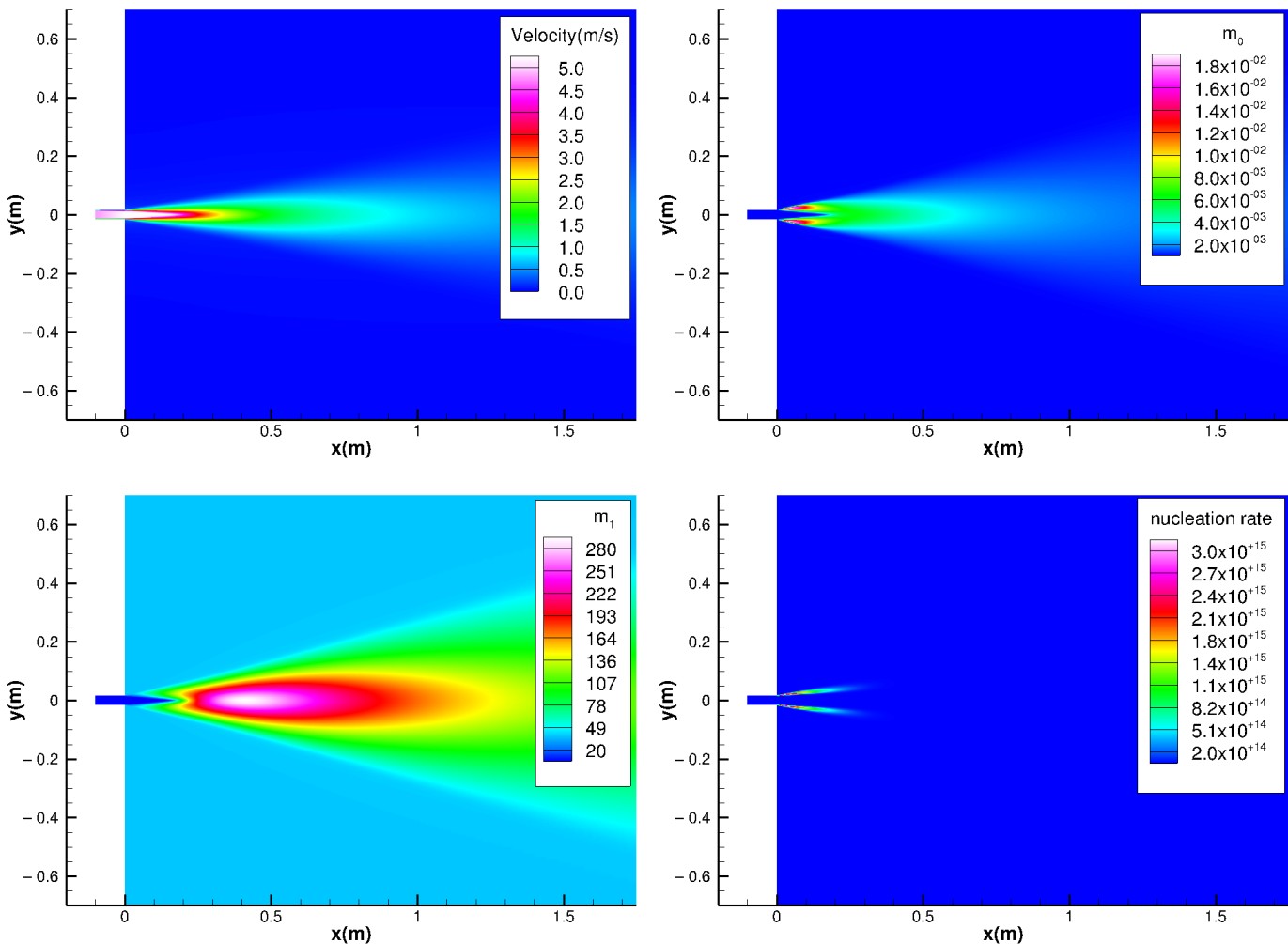

**Figure 2.** No polluted flow field data in 2004.

As Figure 3 shows, in three years, the $m_0$ under nonpolluted conditions on each sampling line is greater than the $m_0$ under polluted conditions. Seipenbusch and Yu once discovered this phenomenon and gave a qualitative explanation [18]. Seipenbusch's work focuses on measured data. The work only analyzes the result of quantity concentration but does not analyze how the various processes have an integrated effect.

The initial value in Table 2, $m_0$ under no polluted conditions is already more significant than that under polluted conditions. The right end of Figure 3 is the value at the boundary, that is, the environmental value. Except for the apparent difference in 2004, the two lines overlapped in the other two years. However, at the left end of the figure, this difference is significantly larger than the difference in the environment. Vehicle emissions will promote the difference in background aerosols. That is, the number concentration under non-polluting conditions is greater than the number concentration under pollution conditions.

In the process of aerosol evolution, nucleation and coagulation will affect the number concentration of particulate matter. Nucleation will increase the number concentration of particulate matter, and aggregation will reduce particulate matter concentration. As shown in Figure 4, on the sampling line, x = 0.1 m and x = 0.25 m, the nucleation rate under the nonpolluted condition is greater than the nucleation rate under the polluted condition. On the sampling line, x = 0.5 m and x = 0.75 m, the nucleation rate in the second half of the nonpolluted situation is greater than the nucleation rate in the polluted condition, and the opposite is the case in the first half. Since the ordinate axis is a logarithmic coordinate, at

x = 0.1 m and x = 0.5 m, the part where no pollution exceeds the pollution is much larger than the difference between x = 0.5 m and x = 0.7 m. The situation has been the same for three years. The lower nucleation rate under the polluted condition is the direct cause of the lower $m_0$ under the polluted condition. As for why the nucleation rate is lower under polluted conditions, it needs to be analyzed by other physical quantities.

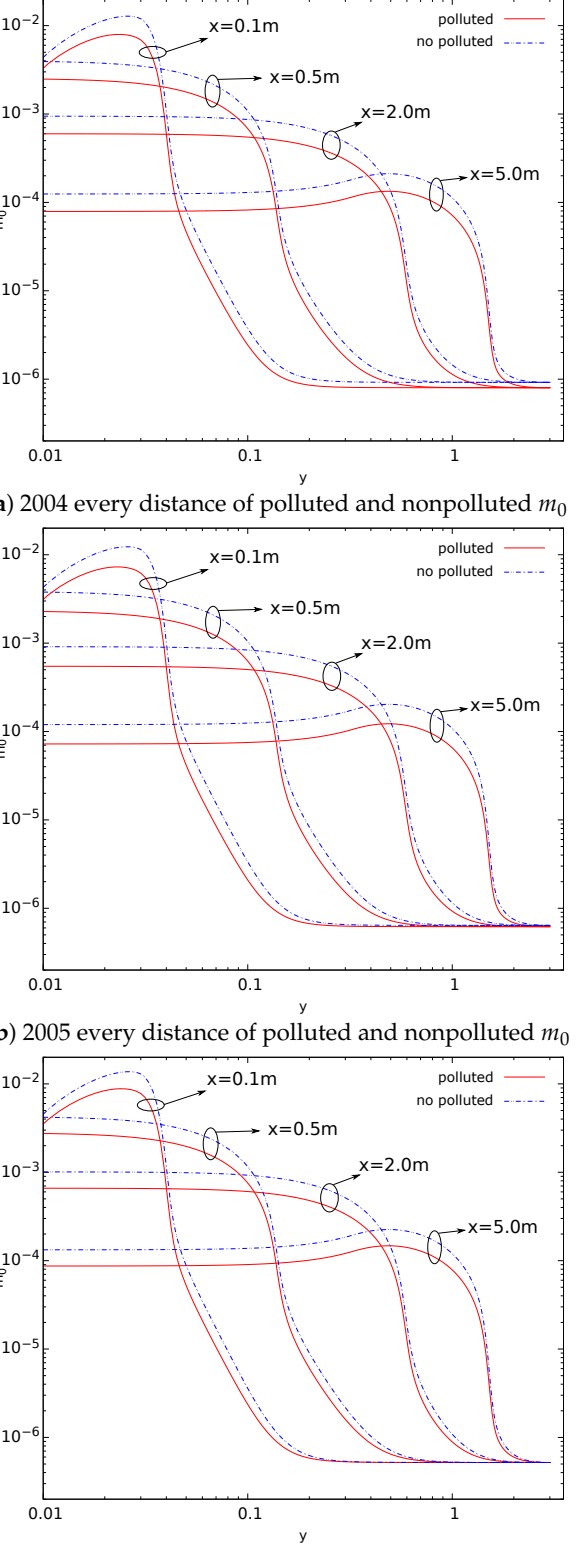

(**a**) 2004 every distance of polluted and nonpolluted $m_0$

(**b**) 2005 every distance of polluted and nonpolluted $m_0$

(**c**) 2006 every distance of polluted and nonpolluted $m_0$

**Figure 3.** Every distance of polluted and nonpolluted $m_0$ over three years.

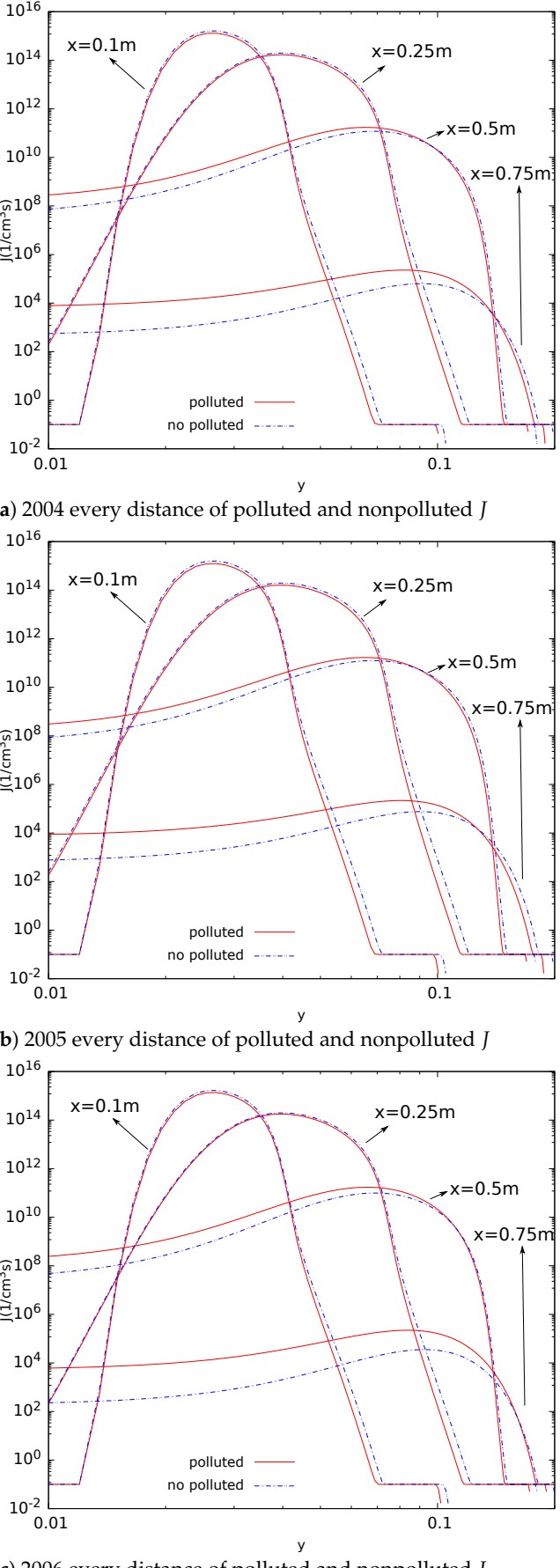

(**a**) 2004 every distance of polluted and nonpolluted *J*

(**b**) 2005 every distance of polluted and nonpolluted *J*

(**c**) 2006 every distance of polluted and nonpolluted *J*

**Figure 4.** Every distance of polluted and nonpolluted *J* in three years.

The value of each moment in the moment method contains the physical properties of the particles, such as surface area, volume, and diameter, and these physical quantities are not independent of each other. For the same volume of particle population, the smaller the particle size, the larger the total surface area of the particle population. For the particle population of the same volume, the larger the particle size, the fewer the particles. The correlation between these variables can be used to infer the mechanism of their evolution. As shown in Figure 5, it is the volume average particle diameter of the particles, defined as $(6m_1/\pi m_0)$. It can be seen that the particle size under the pollution condition in three years is always larger than the particle size under the no pollution condition. A large particle size means a larger surface area, which is suitable for condensation. Under polluted conditions, water molecules and sulfuric acid molecules are more involved in condensation, so the critical molecular clusters involved in nucleation are reduced, resulting in a lower nucleation rate.

This conclusion needs to be corroborated. First, introduce another physical quantity. In the log-normal theory, the geometric mean volume for particles is:

$$v_g = \frac{m_1^2}{m_0^{3/2} m_2^{1/2}} \tag{22}$$

From the area formula of the sphere, $s = \pi d^2$ and volume formula of the sphere, $v_g = (\pi/6)d^3$ the relationship between area and volume is:

$$S = (36\pi)^{1/3} v^{2/3} \tag{23}$$

Substituting Equation (22) into Equation (23) is the surface area of the particles, and multiplying by the number of particles, $M_0$, we can get the total particle surface area. Figure 6 compares the total particle surface area in the case of pollution and no pollution.

Under pollution conditions, the volume average diameter is larger, but the number concentration is minor. If the total volume of particulate matter under polluted and no polluted conditions is equal, then the smaller the number concentration, the smaller the particulate matters's total area. However, it can be seen from Figure 6 that the total area of particulate matter is larger under polluted conditions. This means that the total volume of particulate matter in a polluted condition is greater than the total volume of particulate matter in a no polluted condition. This is confirmed in the data of $m_1$ in Figure 7. At the same time, it should be noted that the nucleation rate is higher under no polluted conditions. The higher nucleation rate increases the total volume while increasing the total area. The total volume of particulate matter reflects the mass concentration of particulate matter, which is an index used to measure the degree of pollution. On this index, the total volume of particulate matter under no polluted conditions is always less than the total volume under polluted conditions.

In Figure 6, at the sampling line close to the exhaust tailpipe, such as x = 0.1 m, the total area of particulate matter under polluted conditions is more significant than that under no polluted conditions. This tendency diminishes as the sampling line moves away from the exhaust tailpipe. On the x = 5.0 m sampling line, the total area of particulate matter under no polluted condition exceeds that under polluted conditions. However, on the sampling line x = 0.5 m, the total area under polluted condition is still larger than that under no polluted conditions at the right end of the horizontal axis (that is, the amount in the environment).

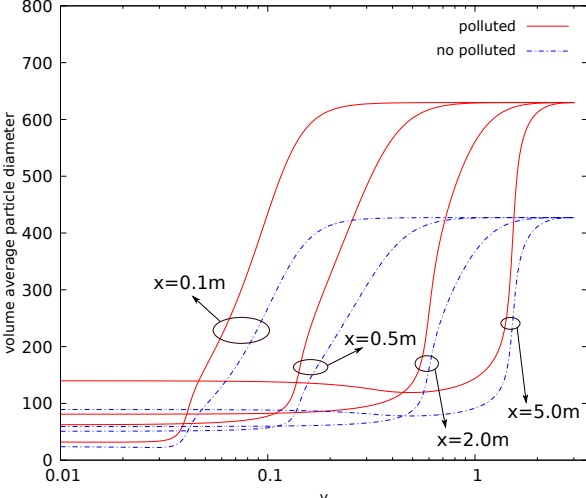

(**a**) 2004 every distance of polluted and nonpolluted volume average particle diameter

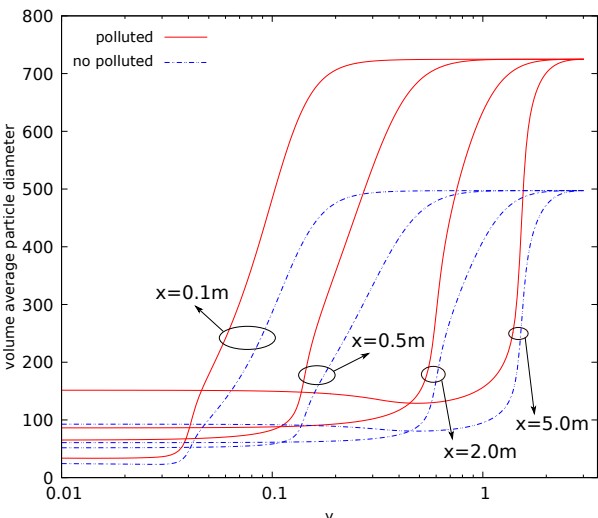

(**b**) 2005 every distance of polluted and nonpolluted volume average particle diameter

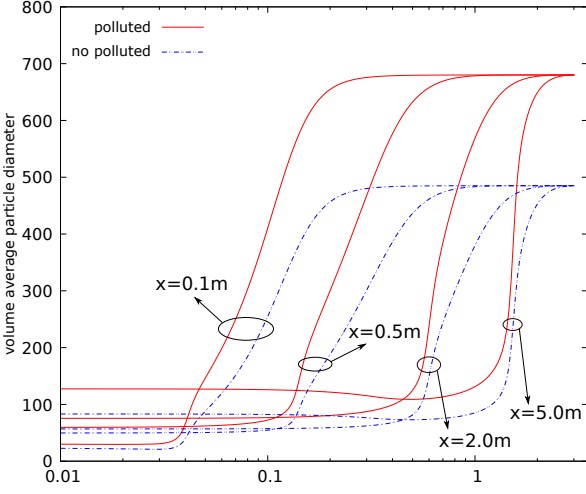

(**c**) 2006 every distance of polluted and nonpolluted volume average particle diameter

**Figure 5.** Status of volume average particle diameter over three different years.

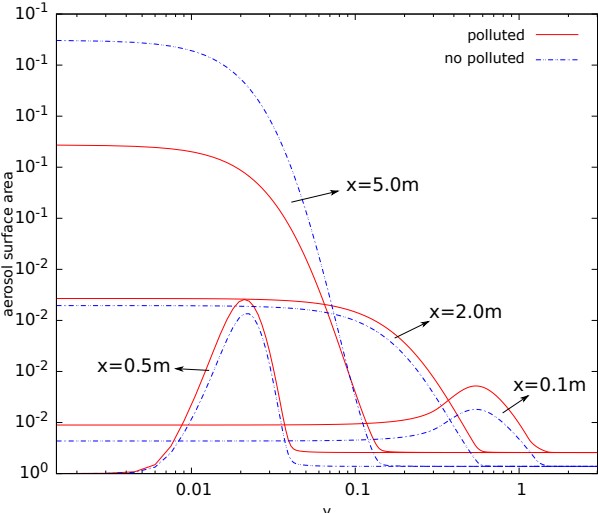

(**a**) 2004 every distance of polluted and nonpolluted aerosol surface area

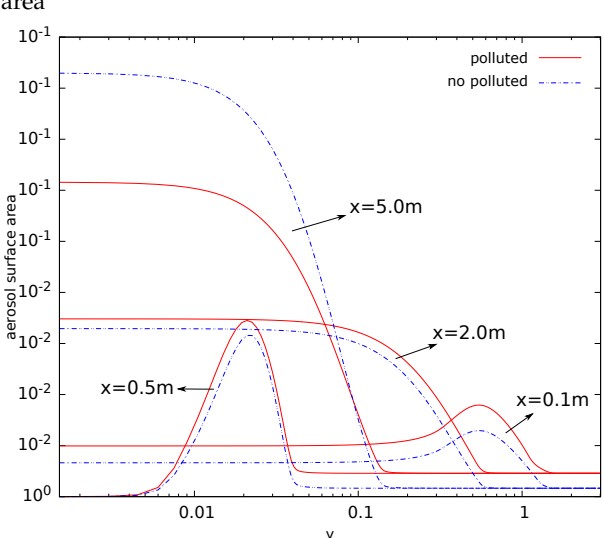

(**b**) 2005 every distance of polluted and nonpolluted aerosol surface area

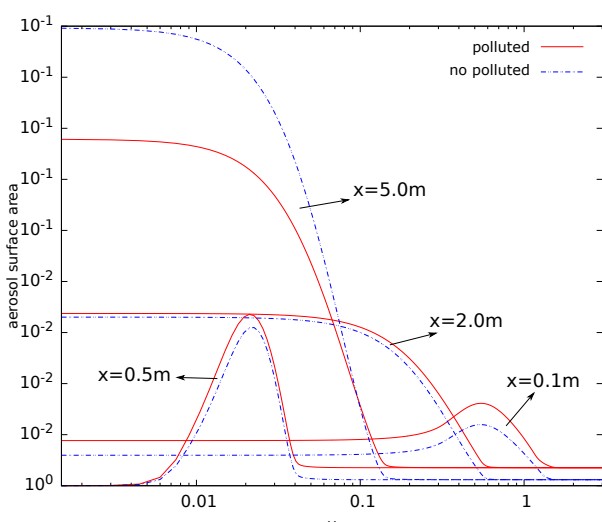

(**c**) 2006 every distance of polluted and nonpolluted aerosol surface area

**Figure 6.** Status of aerosol surface area over three different years.

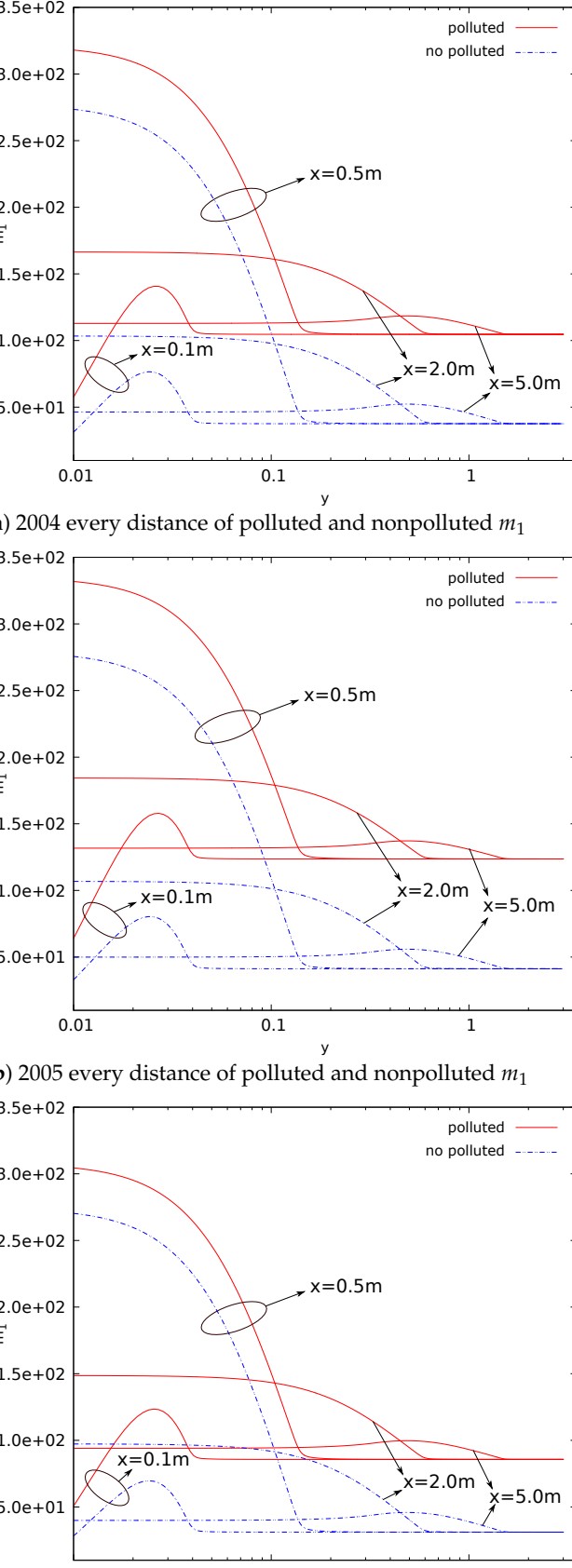

(**a**) 2004 every distance of polluted and nonpolluted $m_1$

(**b**) 2005 every distance of polluted and nonpolluted $m_1$

(**c**) 2006 every distance of polluted and nonpolluted $m_1$

**Figure 7.** Status of $m_1$ over three years.

Compared with the nucleation rate in Figure 4, it can be seen that the nucleation process is mainly near the sampling line x = 0.1 m and x = 0.25 m. The nucleation process results in a greater number concentration in no polluted condition. However, in this area, the total surface area of the particles under no polluted conditions is less than that under polluted conditions. When the sampling line is far away from the exhaust tailpipe, in Figure 6, the number concentration under no polluted condition is always greater than the concentration under the polluted condition. However, the difference between the total area of the particle population under the no polluted condition and the total area of the particle population under the polluted condition is reversed. This shows that the condensation process is mainly carried out in the area from x = 2.0 m to x = 5.0 m. The condensation process increases the surface area of the particles. This leads to an overshoot of the total particle area in no polluted conditions shown in Figure 6. Coagulation mainly occurs in the area after x = 5.0 m, which causes the total surface area of the particulate matter under no polluted conditions to be consistent with the environment and smaller than the total surface area of the particulate matter under the pollution conditions.

## 5. Conclusions

The coupling of the NS equation and the PBE is performed to study the effect of background aerosol on its evolution. The flow field simulation is implemented with SIMPLEC; PBE is solved by TEMOM. Coagulation, condensation, and nucleation are considered. Three years of data were used in the simulation to ensure the generality of the conclusions. According to the observed phenomenon, the number and concentration of particulate matter in the pollution condition are smaller. The reason and mechanism behind it are studied. The following conclusions can be drawn:

1.  Compared with nonpolluted conditions (low concentration of background particles), the nucleation rate in polluted conditions will be reduced. In addition, the number concentration of particles will also be reduced. This phenomenon makes number concentration unable to reflect the degree of pollution. The decrease in nucleation rate makes the concentration of critical sulfuric acid molecular clusters higher in the flow field (this is limited to a relatively small area). This is a disadvantage in certain operations that remove contaminants;

2.  The evolution of aerosols mainly depends on the nucleation rate. In addition, the nucleation rate has a strong regional distribution (that is, the spatial distribution of the nucleation rate is limited to a small range). Therefore, the source of controlling the pollutant is the key. Compared with the core area of nucleation, other regions have little impact on the generation and evolution of particulate matter;

3.  The presence of background particulate matter will slow down the nucleation process of aerosol. The mass concentration of particulate matter in the polluted condition is higher than that in the nonpolluted condition. Therefore, the slowing effect of particulate matter on the nucleation process of aerosol will be more obvious in pollution conditions;

4.  In the process of aerosol evolution, the independence of mass concentration and number concentration has guiding significance in various occasions involving the control of aerosol concentration. In this paper, the evolution mechanism behind this phenomenon is obtained by the simulation method. This is conducive to the use of this phenomenon. For example, it is necessary to confirm which factors effectively control the pollution situation; filtering to reduce the mass concentration of pollutants in order to reduce the number concentration is invalid;

5.  The significant change regions of nucleation, condensation , and coagulation are different. The nucleation process is spatially closer to the source of pollution. The nucleation process also provides more particles for the condensation and coagulation processes. Condensation comes next, and coagulation comes last.

**Author Contributions:** Y.L. devised the project. M.Y. the main conceptual ideas and proof outline. C.T. worked out the technical details, and performed the numerical calculations for the suggested simulation. T.W. verified the numerical results. Conceptualization, Y.L.; methodology, M.Y.; software, C.T.; writing original draft, C.T.; writing review and editing, Y.L.; visualization, T.W. All authors have read and agreed to the published version of the manuscript.

**Funding:** This research received no external funding.

**Institutional Review Board Statement:** Not applicable.

**Informed Consent Statement:** Not applicable.

**Data Availability Statement:** Not applicable.

**Conflicts of Interest:** The authors declare no conflict of interest.

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
