# Peer review of "Simulation of Aerosol Evolution within Background Pollution for Nucleated Vehicle Exhaust via TEMOM"

_applsci, doi:10.3390/app11104552_

Round 1

Reviewer 1 Report

The paper is generally good and interesting. However, the english language needs a thorough workover and improvement. You should use a professional translator with good knowledge on the topic to read through and correct it.

I just give a few examples here:

Already from line 91 and onwards the symbols are too small, and difficult to read.  It seems they are in "lowercase".

In equation (14) the symbol v_c is not defined.

In line 186, "Axisymmetric" - should be with small "a".

In line 220: "six background particles" - do you mean "particle sizes"?

In line 231: "nucleation rate has no value" - do you mean the  value is zero?

In line 239: do you mean "in two-dimensional figures" ? (plots)

Line 250: "Seipenbusch mainly researching the measurement data." This is an incomplete sentence.

At some places the sentences are ambiguous. Take one example in line 345; do you mean "Polluted background particles" or "Polluting background particles"? That is; do you mean that background particles are already polluted, or do they pollute and slow down the nucleation process?

Good luck with correcting and updating.

Author Response

The file is attached.

Reviewer 2 Report

The file is attached.

Author Response

The file is attached.

Round 2

Reviewer 2 Report

Paper "Simulation of aerosol evolution within background pollution for nucleated vehicle exhaust via TEMOM" can be accepted in present form with some corrections. Line 23, 24. Double «Therefore…». Correct, please. Line 63. There is no space in the sentence. Line 117. There is no space in the sentence. Line 143. There is no space in the sentence. Line 169: “moment transport equation”. Starts from the small letter. Line 170. There is no space in the sentence. Line 177, 178: Starts from the small letter. Line 198: Starts from the small letter.